# An Efficient, Lightweight, Tiny 2D-CNN Ensemble Model to Detect Cardiomegaly in Heart CT Images

**DOI:** 10.3390/jpm13091338

**Published:** 2023-08-30

**Authors:** Bhanu Prakash Doppala, Ali Al Bataineh, Bandi Vamsi

**Affiliations:** 1Data Analytics, Generation Australia, 88 Phillip St, Sydney, NSW 2000, Australia; bhanu.doppala@generation.org; 2Artificial Intelligence Center, Norwich University, Northfield, VT 05663, USA; 3Department of Computer Science—Artificial Intelligence & Data Science, Madanapalle Institute of Technology & Science, Madanapalle 517325, India; vamsib111@gmail.com

**Keywords:** cardiomegaly, convolutional neural networks, U-Net, heart disease detection, lightweight neural network, Tiny-ML

## Abstract

Cardiomegaly is a significant global health concern, especially in developing nations. Although advanced clinical care is available for newly diagnosed patients, many in resource-limited regions face late diagnoses and consequent increased mortality. This challenge is accentuated by a scarcity of radiography equipment and radiologists. Hence, we propose the development of a computer-aided diagnostic (CAD) system, specifically a lightweight, tiny 2D-CNN ensemble model, to facilitate early detection and, potentially, reduce mortality rates. Deep learning, with its subset of convolutional neural networks (CNN), has shown potential in visual applications, especially in medical image diagnosis. However, traditional deep CNNs often face compatibility issues with object-oriented human factor technology. Our proposed model aims to bridge this gap. Using CT scan images sourced from the Mendeley data center, our tiny 2D-CNN ensemble learning model achieved an accuracy of 96.32%, offering a promising tool for efficient and accurate cardiomegaly detection.

## 1. Introduction

Cardiovascular diseases (CVDs) consistently present a major global health challenge. As of 2019, they were responsible for an estimated 17.9 million deaths, comprising roughly 32% of all global fatalities. Among the array of conditions under the CVD umbrella, cardiac hypertrophy stands out, characterized by the enlargement or thickening of heart muscles. Such changes, often precipitated by conditions like hypertension or chronic cardiovascular diseases, have been shown to predispose individuals to life-threatening events like heart failure and sudden cardiac episodes [1,2].

Diagnostic strategies for detecting cardiac abnormalities, including hypertrophy, have evolved markedly over the years. Advanced imaging techniques, such as X-rays, CT scans, and MRIs, have become standard. In particular, the cardiothoracic ratio, derived from X-ray images, serves as an important diagnostic metric, with ratios exceeding 50% indicating potential hypertrophy, contingent upon accurate interpretation [3,4,5].

However, with the advancement in imaging technologies comes the challenge of managing and interpreting vast amounts of medical data. Despite the proliferation of imaging modalities, there is a pronounced shortage of specialized radiologists. This disparity has highlighted the potential of artificial intelligence (AI), specifically deep learning (DL), to fill the gap. DL, especially its subset known as convolutional neural networks (CNNs), excels in analyzing medical images, acting as an effective tool for pattern recognition and diagnosis [6,7,8,9,10]. DL applications on medical data are visualized in Figure 1.

However, the computational demands of traditional DL pose significant barriers, particularly in resource-limited settings. In response, tiny machine learning (TML) emerges as a potential game changer [14,15]. TML encapsulates the capabilities of AI within small, cost-effective computational units known as microcontrollers, presenting an innovative solution for on-the-spot, resource-efficient diagnostics [16,17]. The workflow of the TML is depicted in Figure 2, where a dataset is trained using an ML model, and that will be passed to TinyML based on the new image passed; this model predicts the presence of the disease.

Guided by this backdrop, our study focuses on the integration of DL and TML, striving to detect cardiac hypertrophy from heart CT images. By leveraging a combination of a lightweight neural network and ensemble learning, we aimed to craft a solution that not only provides accurate diagnostic insights but also overcomes the computational limitations of traditional methodologies. Here is a summary of our work:(a)We started with a two-way labeled image for heart enlargement segmentation.(b)We applied several analysis methods to the image input in heart enlargement analysis. After a comprehensive literature review, we identified several limitations of these methods, such as high processing power and GPU requirements, as well as long training times.(c)We proposed a hybrid framework that combines deep learning and machine learning models. This approach was able to address some of these limitations and produce stable results.

The rest of this paper is organized as follows: Section 2 discusses related work. Section 3 explains the ML methods and models. Section 4 describes the results of the proposed model, followed by a discussion in Section 5. Finally, the conclusion and future scope are elaborated in Section 6.

## 2. Related Work

This section reviews various research work conducted on medical imaging using deep learning techniques.

Kim H-E et al. developed a CAD model related to deep CNN for technical TB analysis. They leveraged the benefits of transfer learning to achieve notable TB analysis outcomes with an area under the curve of 0.96, 0.93, and 0.88 on three real-world, large-scale chest X-ray (CXR) records [18].

Han X’s suggested that the DCNN approach achieved a mean absolute error (MAE) of less than 85 HU for 13 out of 18 test participants. The overall average MAE was 84.8 ± 17.3 HU, significantly lower than the overall MAE of 94.5 ± 17.8 HU achieved by the atlas-based method [19].

Xuan Y, et al. proposed a U-Net network (DC-U-Net) combined with dilated convolution. The DC U-Net’s image segmentation performance was closer to the ground truth compared to other methods, achieving an intersection over onion (IoU) of 0.9627 and a Dice coefficient of 0.9743 [20].

Hooda et al. introduced an encoder−decoder network specially designed for lung field segmentation. The suggested network was tested, trained, and evaluated on freely available high-quality datasets, demonstrating superior performance compared to state-of-the-art methods, with an accuracy of 98.73% and an overlap of 95.10% [21,22].

Park S et al. proposed a deep learning model that significantly outperformed other medical classifiers in image-based classification (98.3% and 81.4%) and lesion-wise localization (98.5% and 78.1%). The implementation of the algorithm led to significant improvements in image-related division and lesion-based localization across all three physician groups [23].

The weighted classifier proposed by Hashmi et al. outperformed all other methods, achieving an examination accuracy of 98.43% and an area under the curve of 99.76% [24].

Antani S et al. evaluated a custom CNN and a well-known pre-trained CNN on a large-scale publicly available X-ray dataset. A grouped ensemble of these priority-based, three-valued methods showed promising results [25].

Irvin J et al. proposed CheXNeXt, a CNN capable of identifying 14 different diseases simultaneously. CheXNeXt outperformed in 11 pathologies but underperformed in three [26].

Petrick N et al. proposed a CNN architecture and texture feature settings, achieving an area under the ROC of 87% [27].

Vamsi et al. used a lightweight CNN with a Random Forest classifier to identify stroke regions of the brain, achieving an accuracy of 97.81% [28].

From the above work [18,19,20,21,22,23,24,25,26,27,28], it is observed that most researchers have proposed segmentation work for identifying regions of cardiomegaly and other diseases using heavyweight convolution networks. Existing lightweight work [28] has primarily used specific ML algorithms for extracting pixel values from data frames. To address these limitations, we propose an ensemble-based voting classifier model for extracting pixel values from data frames to identify enlarged regions more effectively.

## 3. Methods

This section covers the several deep learning methods, their functioning procedures, augmentation and data preprocessing approaches, and the metadata utilized in the study.

### 3.1. Data Augmentation

The primary issue at hand is that the labeling of training sets often takes precedence over proper model fitting. This problem can be mitigated using data augmentation techniques. Research on augmentation strategies, including traditional, spatial, and intensity-based modifications, is prevalent in this field [29]. The latest image generation technologies offer various types of data augmentation, some of which leverage online community resources. Data augmentation is not only applicable in clinical fields but is also utilized in areas such as medical CT scans and natural image processing, as supported by extensive literature. For this research, we employed image inversion in both horizontal and vertical orientations. The results of various comparisons are presented in Figure 3.

### 3.2. U-Net Model with CNN

This is mostly used for semantic image segmentation and it is a pre-defined architecture with any CNN model as the basis architecture. Downsampling and upsampling are the two steps of the U-Net algorithm. Encoding and decoding are other names for them. The factors located in the picture are retrieved through neural convolutional layers during the downsampling phase. The image is formed depending on retrieved factors during the upsampling or decoding phase. The input layer convolution I(l)  with a filter image F(l) together predicts the output layer O(l) of the 2D-CNN and is represented by Equation (1).
(1)$$O\left(n_{1},n_{2}\right)=\sum_{i_{1}=0}^{M-1}\sum_{i_{2}=0}^{N-1}I\left(i_{1},i_{2}\right)\;F(n_{1}-i_{1},n_{2}-i_{2})$$
where M represents the regions of max-pooling layers, N denotes the number of convolutions, $n_{1},n_{2}$ represents the dimensions, and $i_{1},i_{2}$ represents the kernels of the respective convolutions. Figure 4 depicts the U-Net design in detail and demonstrates that the model is given an input image. The input image’s dimensions are 224 × 224 × 3. In the working procedure of downsampling, the first layer of the model includes 16 convolutions with a stride value of ‘2’ for the size 3 × 3, and this process is continued for 32 and 64 convolutions of the same. In the initial layer, for the given input images, the max-pooling technique is applied, and the resulting image has dimensions of 112 × 112 × 32. The input image undergoes a similar method until it reaches the dimensions of 28 × 28 × 128, with intermediate reductions of 56 × 56 × 64 and 28 × 28 × 128 as shown in Figure 4.

The downsampling method’s output is passed to the initial layer of upsampling, which decodes image dimensions 28 × 28 × 128 into a convolution of 56 × 56 × 64. The same method is repeated until the image has 224 × 224 × 3 dimensions. The desired filter is a final picture with dimensions of 224 × 224 × 3 that is utilized to segment the input image, as depicted in Figure 5.

### 3.3. Architecture of VGG-16

The visual geometric group (VGG) presented a network with 16 layers, as depicted in Figure 5. It has trainable parameters in each of the 16 levels with 5 blocks each, as well as layers like Max pool in the middle. It features two continuous convolution layers before the max pooling. These are classified into three blocks of three convolution layers, each labeled as having a maximum pooling layer. VGG-16 takes a 224 × 224 × 3 input image as input. There are a few tiny receptive convolution kernels with 3 × 3 dimensions and a stride of 1. The images are processed through a stack of these kernels to train the parameters in each convolution layer. It also has max-pool layers that are 2 × 2 in size and a stride value of 2. The architecture of VGG-16 is depicted in Figure 6.

### 3.4. Dataset

The dataset for this study, related to cardiomegaly, was collected from the Mendeley Data Center [30]. A CT scan, also known as a CAT scan, is a diagnostic medical imaging procedure that provides numerous images of the body’s interior, similar to standard X-rays. For this study, DICOM images of 20 subjects were gathered, with 11 diagnosed with cardiomegaly and nine considered healthy. This dataset was approved by a multispecialty hospital in India. The chosen dataset provided a sufficient number of samples to conduct the study. It is often challenging to find an open-source dataset specific to cardiomegaly, making this one particularly valuable.

### 3.5. Ensemble Model

The proposed method employs an ensemble voting classifier model to improve classification performance. This model combines three classifiers (Naïve Bayes, Random Forest, XGBoost) and is primarily used to identify the enlarged region of the heart. The top convolutions of the VGG-16 architecture produce frames of 8 × 8 dimensions. The pixel data generated from these frames are passed to the proposed ensemble classifier to identify the heart’s enlargement, as represented in Figure 7.

#### 3.5.1. Gaussian Naïve Bayes

It is a conditional probability theorem that applies to conditional probabilities. If one thing else has presently happened, relative probability is actually the possibility that one thing will happen. Therefore, relative probability may figure out the chance of an occasion utilizing anticipation—probability represented in Equation (2).
(2)$$P\left(A|B\right)=\frac{P\left(B|A\right)*P\left(A\right)}{P\left(B\right)}$$

P(A): The likelihood of hypothesis H is correct. Prior probability is the term for this.

P(B): The likelihood of the evidence.

P(A|B): If the hypothesis is correct, the evidence’s probability.

P(B|A): If the evidence is proper, the hypothesis’s likelihood.

The absolute most preferred method for using Naïve Bayes to real-valued information is actually to presume a Gaussian circulation. Naturally, various other features could be made use of to determine information distributions. Still, the Gaussian (or normal) circulation is actually the simplest to partner with given that it merely demands you to determine the method as well as standard deviation coming from your instruction data. It is easy to determine each input variable’s method and standard deviation (x) for every class value using Equation (3).
(3)$$\mathrm{mean}\left(x\right)=\frac{1}{n}*\mathrm{sum}\left(x\right)\;$$

The value of an input variable is x, and the number of instances in your training data is n. Equation (4) can be used to compute the standard deviation:(4)$$\mathrm{standard}\;\mathrm{deviation}\left(x\right)=\mathrm{sqrt}\left(\frac{1}{n}*\mathrm{sum}\left(\mathrm{xi}-\mathrm{mean}{\left(x\right)}^{2}\right)\right)$$

Standard deviation is the straight origin of the average squared difference of each worth of x coming from the mean value of x, where ‘n’ is actually the number of circumstances, sqrt() is the square root functionality, sum() is actually the amount function, xi is actually the certain market value of the x variable for the ith circumstances, and mean(x) is the mean value of the x variable.

A Gaussian Naïve Bayes model was employed to produce predictions. The probability density function (PDF) is used to calculate the probabilities for new x values using Equation (5).
(5)$$\mathrm{pdf}\left(x,\mathrm{mean},\;\mathrm{sd}\right)=\left(\frac{1}{\sqrt{2*\mathrm{PI}*\mathrm{sd}}}*\left(-\frac{x-{\mathrm{mean}}^{2}}{2{*\mathrm{sd}}^{2}}\right)\right)$$

#### 3.5.2. Random Forest (RF)

By using RF, the final result is calculated through a group of basic trees. In categorization problems, the most desired class is generated through an ensemble of simple tree votes. Their replies are averaged to provide a dependent variable estimate in the regression problem. The usage of tree ensembles can enhance prediction accuracy significantly. The response of each tree is decided by a portion of the original dataset’s predicted values, which are generated separately (with replacement) and spread uniformly across the forest. $log^{2}M+1$, where ‘M’ is defined as the number of inputs, gives the optimal subset of predictor variables. From a group of simple trees along with random predictor factors, the RF technique produces a limit concern, which shows the proportion of the total count of votes under the correct class to that of the class across the total votes of different classes in DV. This makes it easier to make predictions and assign a level of confidence to those forecasts. For a Random Forest, the mean-square error is obtained by Equation (6).
(6)$$\mathrm{mean}\;\mathrm{error}={\left(\mathrm{observed}-\mathrm{tree}\;\mathrm{response}\right)}^{2}$$

The average of the tree projections is meant to be the Random Forest forecasts represented in Equation (7).
(7)$$\mathrm{Random}\;\mathrm{forest}\;\mathrm{prediction}\;s=\frac{1}{K}\sum_{k=1}^{K}K^{\mathrm{th}}\mathrm{tree}\;\mathrm{response}$$

#### 3.5.3. XGBoost

XGBoost (eXtreme Gradient Boosting) is a famous technique that boosts the efficiency and speed of tree-based (sequential selection plants) artificial intelligence protocols. XGBoost was made through Tianqi Chen and was first sustained due to the dispersed (rich) artificial intelligence community (DMLC). Let us consider that we have a dataset with n examples. We will use the letter I to represent each example. XGBoost builds trees using the loss function, which minimizes the following value using Equations (8) and (9).
(8)$$ℒ\left(\emptyset \right)=\underset{i}{\sum }ı\left({\hat{y}}_{i},y_{i}\right)+\underset{k}{\sum }\Omega \left(f_{k}\right)$$
where
(9)$$\Omega \left(f\right)=\;\Upsilon T+\frac{1}{2}⋋{\left\|w\right\|}^{2}$$

The loss function calculates pseudo residuals of the predicted value ${\hat{y}}_{I}$ and true value $y_{I\;}$ in each leaf in the first section, while the second part is separated into two portions, as shown above. The regularization termed lambda that is aimed to decrease the prediction’s sensitivity to specific observations is found in the final portion of the omega formula, and ‘w’ defines weights of leaf that can also be thought of as a leaf resultant value. Furthermore, ‘T’ specifies the number of leaf nodes in a tree, whereas gamma is the user-defined pruning penalty. The goal is to find the leaf’s best output value to keep the overall equation minimum. The preceding prediction is constantly equal to the i−1  forecast, along with the resultant value from the $i^{\mathrm{th}\;}$ tree because that starts with a value of $y_{0}$ is calculated by Equations (10) and (11).
(10)$${ℒ}^{\left(t\right)}=\overset{n}{\underset{i=1}{\sum }}ı\left(y_{i,}{\hat{y}}_{i}{}^{\left(t-1\right)}+f_{t}\left(x_{i}\right)\right)+\Omega \left(f_{t}\right)$$

We use $g_{I}\;$ to represent the loss function’s first derivative because it is correlated to the gradient, and here $h_{I}\;$ to define the other derivative because it is related to the Hessian.
(11)$${ℒ}^{\left(t\right)}=\overset{n}{\underset{i=1}{\sum }}\left[ı\left(y_{i,}{\hat{y}}_{i}{}^{\left(t-1\right)}\right)+g_{i}f_{t}\left(x_{i}\right)+\frac{1}{2}h_{i}f_{t}^{2}\left(x_{i}\right)\right]+\Omega \left(f_{t}\right)$$

#### 3.5.4. Proposed Voting Classifier Mechanism

The voting classifier is one of the ensemble algorithm models. In regression, the voting process usually generates a prediction that is the average of numerous regression models. To create the model, we used NB, RF, SVM, and gradient boosting classifiers. Every model version generates a forecast for each examination circumstance, with the most popular outcome forecast receiving the most votes. If none of the forecasts receives more than half of the votes, we can conclude that the set approach is unlikely to produce a consistent forecast in these conditions. The popularity vote of each classifier $C_{j}$ that is taken into account is used to forecast the class  ^y using Equations (12) and (13).
(12)^y=mod e{C1(x), C2(x),…, Cm(x)}

Computation of majority voting associated with weight $w_{j}$ to the classifier $C_{j}.$
(13)^y=argmaxi∑j=1mwjxA(Cj(x)=i)

$x_{A}$ Characteristic function $\left[C_{j}\left(x\right)=i\in A\right],\;A$ unique label set of a class.

Classifier probability that is predicted using Equation (14).
(14)^y=argmaxi∑j=1mwjpij

## 4. Results

In this section, we present the performance metrics utilized to assess the efficacy of the proposed model in comparison to other models. The evaluations were based on CT images from both enlarged heart patients and healthy heart subjects, with hybrid segmentation employed to identify the enlarged regions. All computations were performed on a machine equipped with an ‘Intel i5 10th generation CPU processor’ and ‘8 GB RAM’.

### 4.1. Data Execution Flow

The data flow of the proposed 2D-CNN model is to identify the enlarged heart area utilizing CT images as input as depicted in Figure 6. The convolutions of sizes 224 × 224 with 64 filters each are used to extract features from the input image of size 224 × 224 × 3. These convolutions are derived from the VGG-16 model’s first layers, which have been trained to detect enlarged heart regions. A feature map with 64 photos is the result of feature extraction. The feature map of every image is flattened, and these images are then concatenated to form a ‘data frame’ having 64 columns. By flattening the labeled image, an additional label column is linked to the data frame. This ensemble model is given to the data frame, including each pixel value of every filter as an input. To forecast the filtered image, the ensemble technique is trained by using the pixel values provided in data frames. The heart CT images are segmented using the anticipated filter. The proposed dataflow model is represented in Figure 8.

### 4.2. Segmentation

The segmentation procedure using anticipated filters is depicted in Figure 9. There are three classes in the filtered image. The network’s intermediator output is a three-class tagged picture. We built the output image with this. The enlarged area is represented by the class containing white pixels, the grey matter is represented by the class having grey pixels, and the other region of the image is represented by the class containing black pixels. The two-class labeled image is required in this case to distinguish the pixel density of the enlarged area. The projected label’s class white area is transferred to create a segmented image through every input image with the affected heart enlargement highlighted in red.

### 4.3. Algorithm Proposed

Step 1: Classify the ‘imagenet’ based on enlarged and healthy heart
Step 2: Train the lightweight model based on data set
Step 3: Initialize the ‘top_conv’ = = ‘FALSE’ by applying ‘imagenet’ with a dimension size as 224 × 224
Step 4: for each layer of VVG-16:
Initialize the layer trainable = = ‘FALSE’
Step 5: Extract the features from top convolutions of VGG-16 architecture
Step 6: Initialize the square size and n: = 1
For each square of rows do:
For each square of columns do:
Extract data pixels of each frame where features with a range of 0 to n − 1
n = n + 1
Step 7: Apply ensemble voting classifier for predicting the enlarged region

### 4.4. Performance Evaluation Metrics

To evaluate the effectiveness and precision of our proposed model, we utilized several performance evaluation metrics. Each metric provides a unique perspective on the model’s performance and, together, they offer a comprehensive assessment [31]. Here are the specific metrics we used:

#### 4.4.1. Accuracy

This measures the percentage of correctly identified cardiomegaly and healthy subjects with respect to the total number of subjects used for the study. It is defined in Equation (15).
(15)$$Accuracy=\frac{True\;\left(Positive+Negative\right)}{Total\;no.\;of\;subjects}$$

#### 4.4.2. Sensitivity or Recall

This measures the percentage of correctly identified cardiomegaly subjects with respect to the sum of correctly identified subjects and truly identified healthy subjects. It is defined in Equation (16).
(16)$$Sensitivity=\frac{True\;Positive}{True\;Positive+False\;Negative}$$

#### 4.4.3. Specificity

This measures the percentage of correctly identified healthy subjects with respect to the sum of misclassified subjects and truly identified healthy subjects. It is defined in Equation (17).
(17)$$Specificity=\frac{True\;Negative}{False\;Positive+True\;Negative}$$

#### 4.4.4. Dice Coefficient

This is an image segmentation calculation metric. It measures the intersection over union (IoU) of every class in the predicted image with the labelled image. Where P,L represent predicted and labelled images. $P_{i\;}$ denotes the pixel of predicted class i, $L_{i}$ represents the pixel of labeled class i, and N denotes the number of pixels in each class. It is defined in Equation (18).
(18)$$DC_{o}\left(P,L\right)=\frac{\sum_{i=1}^{N}2*\frac{\left|P_{i\;}\cap \;L_{i}\right|}{\left|P_{i}\right|+\left|L_{i}\right|}}{N}$$

### 4.5. Performance Evaluation of Proposed Model with Existing Models

Figure 10 exhibits the preprocessed heart CT images belonging to eight random subjects in which two have a healthy, normal heart and six have enlarged heart issues. Figure 11 represents the labeled images where the images contain two classes; white and black. Figure 12 depicts the resultant images of our proposed model.

The Dice coefficient ($DC_{o}$) of each algorithm are displayed in a table with various input images. Likewise, the algorithm’s time is the ratio to the total number of convolutions. The algorithms $DC_{o}$ are related to total number of convolutions. As the capacity to extract features grows as the number of convolutions increases, we applied the ensemble technique to help the VGG-16 model extract more features while reducing the number of convolutions.

The time it took each algorithm to train the input data is shown in Table 1 and the performance of existing models and the proposed model is compared in Table 2. When compared to existing heavyweight model outcomes, our proposed Tiny-2D-CNN-ensemble model achieved an accuracy of 96.32%. Figure 13 compares the proposed model’s AUC metrics to those of existing models.

As per the observations made, the VGG-16 model took 13 h for processing 12 convolutions, five max pooling, and one fully connected layer. U-Net consumed 24 h for processing seven convolutions together, both encoding and decoding, three max pooling, three deconvolutions, and SoftMax with dimensions 224 × 224 × 3. In contrast, the proposed tiny 2D-CNN ensemble model took 5 h of training time for processing top convolutions only, which consumes less training time when compared to heavyweight convolutions.

Table 2 demonstrates the comparison between the models with respect to performance evaluation metrics. As per the comparative results obtained, it is observed that the proposed tiny 2D-CNN ensemble outperformed when compared to the heavyweight VGG-16 and U-Net models in terms of accuracy, sensitivity, and specificity. Table 3 represents the comparative outcome of the heavyweight and proposed tiny 2D-CNN ensemble models in terms of dice values, and its visualization is represented graphically in Figure 14.

The receiver operating characteristic curve is basically a curve used to represent classification thresholds between Sensitivity and Specificity as depicted in Figure 15.

## 5. Discussion

The results derived from our research provide a comprehensive perspective on the capabilities of the proposed tiny 2D-CNN ensemble model, especially when compared to established heavyweight models such as VGG-16 and U-Net. The primary observation to note is the superior accuracy of our tiny 2D-CNN ensemble model, which achieved a commendable 96.32%. When contrasted with heavyweight models, this high accuracy not only suggests an increased model efficiency but also a greater potential for real-world applicability in distinguishing between cardiomegaly and healthy subjects. While accuracy is undeniably critical, it is worth emphasizing that the heightened sensitivity and specificity metrics of the proposed model further accentuate its proficiency. In the domain of medical imaging, where missing an anomaly (false negatives) can be as detrimental as falsely identifying one (false positives), such metrics are invaluable. This makes the proposed tiny 2D-CNN ensemble model a potential frontrunner in clinical settings where precision is paramount.

The segmentation prowess of our method, as illustrated by the Dice coefficient values, offers another lens through which its robustness can be appraised. The derived values indicate that our model provides a more accurate segmentation of enlarged heart areas compared to its heavyweight counterparts. Given the crucial role of accurate segmentation in diagnosis, this achievement underscores the model’s clinical utility. Moreover, the stark difference in training times between the models cannot be overlooked. With the proposed tiny 2D-CNN ensemble model being trained in just 5 h compared to the 13 and 24 h required by VGG-16 and U-Net, respectively, it becomes evident that our model excels in computational efficiency. In practical scenarios where rapid model training and retraining might be essential, such efficiency gains could prove pivotal. This indicates that our model not only outperforms the other models but is also more adaptable to evolving datasets and clinical demands.

However, as with all studies, there are limitations to consider. While our model showed faster training times, its performance in real-time clinical contexts—encompassing inference speed and integration with medical imaging systems—still warrants further investigation. These considerations highlight the importance of ongoing research and refinement of the tiny 2D-CNN ensemble model to ensure its wider clinical acceptance.

## 6. Conclusions and Future Scope

This study highlighted the computational challenges faced by current deep learning models in detecting cardiomegaly. Recognizing the importance of resource efficiency, especially in medical contexts, we introduced the tiny 2D-CNN ensemble model. This model, leveraging the strengths of the VGG-16 architecture and ensemble learning, stands out as a more efficient alternative to traditional heavyweight CNNs. While heavyweight models often entail longer processing times and increased computational costs, our model promises both swift and accurate outcomes.

Recognizing the profound value of benchmarking against established diagnostic measures, a future direction for our research will be a comparative study between the tiny 2D-CNN ensemble model and traditional diagnostic methods such as X-P + echocardiography. This comparison aims to provide a comprehensive understanding of the model’s utility and efficiency in clinical contexts. Additionally, as part of our forward-looking agenda, we envision integrating TinyML with 3D-CNN, which we anticipate will further enhance the detection of cardiomegaly, promising both cost and time efficiencies.

## Figures and Tables

**Figure 1 jpm-13-01338-f001:**
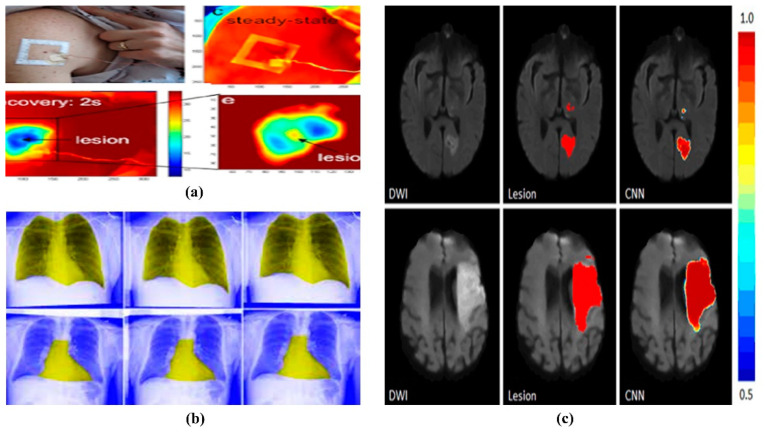
(**a**). Identifying skin cancer lesions [11]. (**b**). Enlarged heart [12]. (**c**). Acute brain stroke [13].

**Figure 2 jpm-13-01338-f002:**
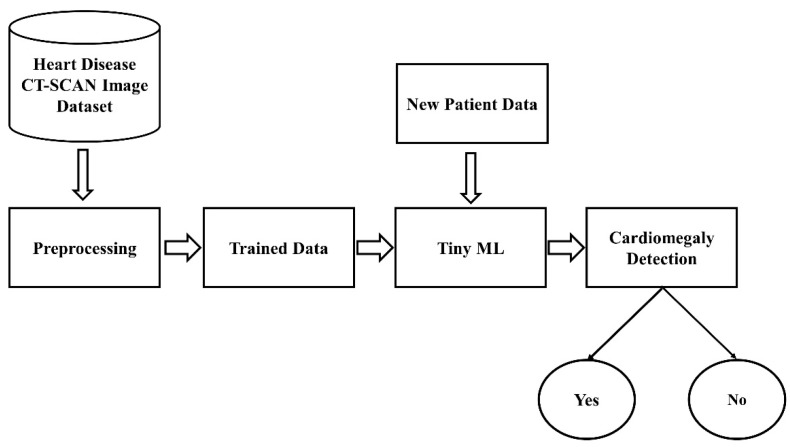
TinyML workflow.

**Figure 3 jpm-13-01338-f003:**
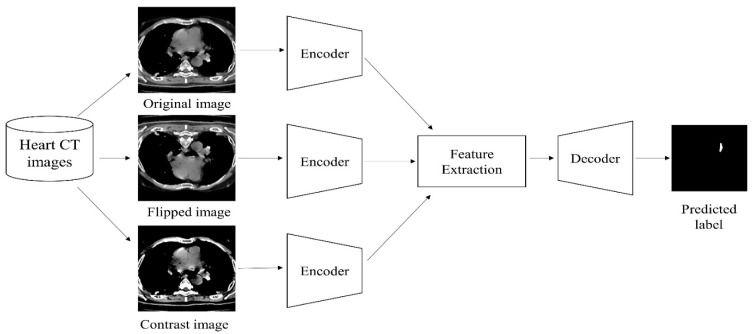
Different forms of data augmentation on chest CT images.

**Figure 4 jpm-13-01338-f004:**
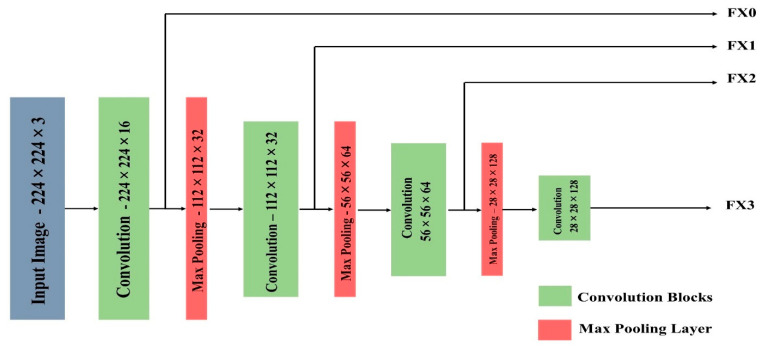
Four convolutional blocks in the U-Net encoder following each convolution: the image dimensions are presented.

**Figure 5 jpm-13-01338-f005:**
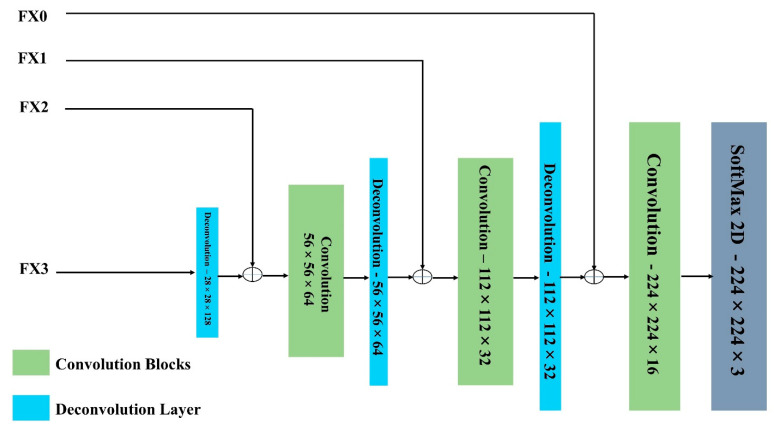
Three deconvolution blocks in the U-Net decoder following each convolution: the image dimensions are presented.

**Figure 6 jpm-13-01338-f006:**
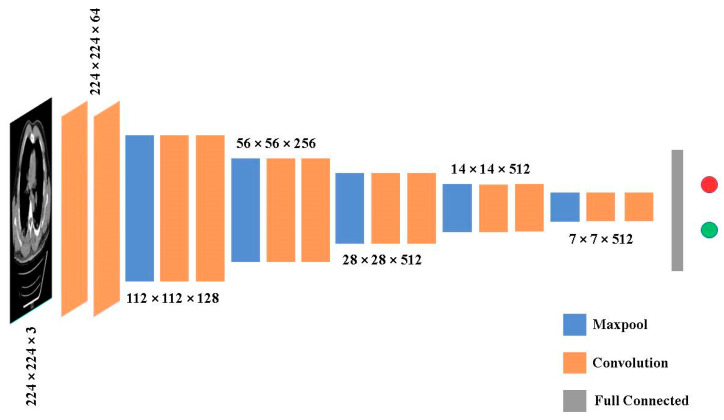
VGG-16 architecture convolution blocks.

**Figure 7 jpm-13-01338-f007:**
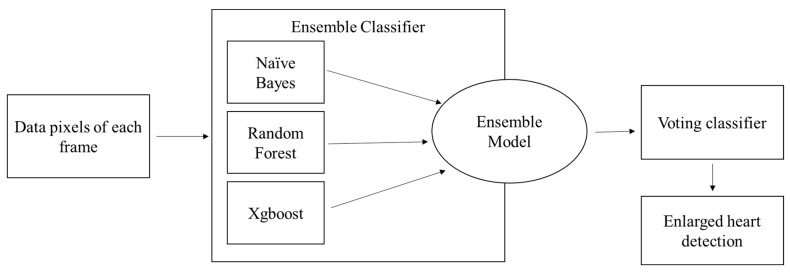
Proposed Ensemble Voting Classifier.

**Figure 8 jpm-13-01338-f008:**
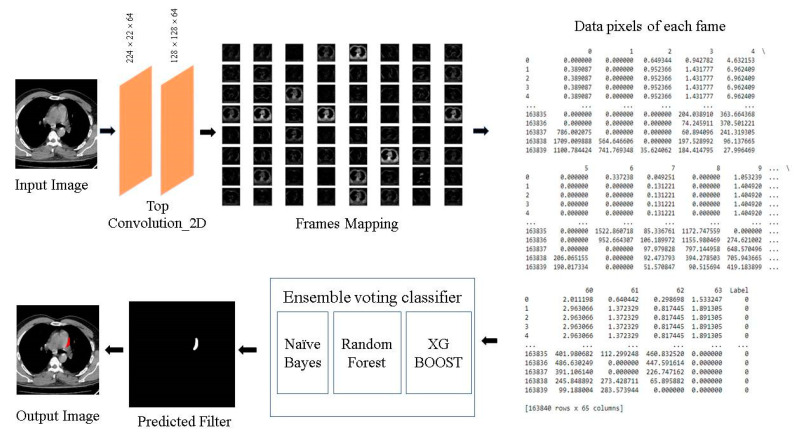
Proposed Models’ Dataflow.

**Figure 9 jpm-13-01338-f009:**
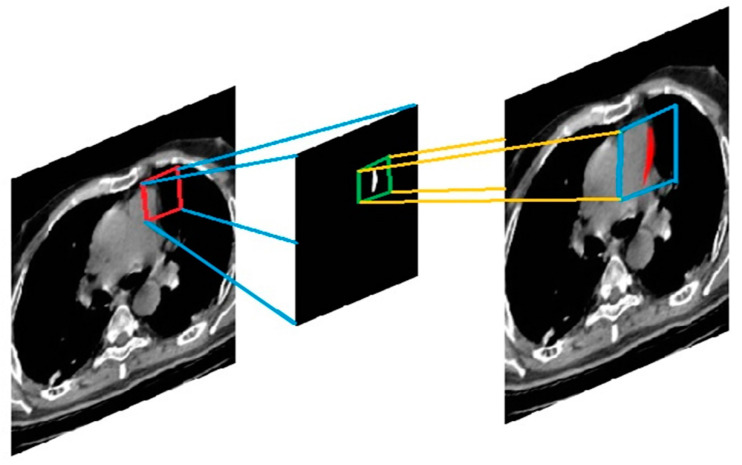
Segmentation process to detect enlarged area.

**Figure 10 jpm-13-01338-f010:**
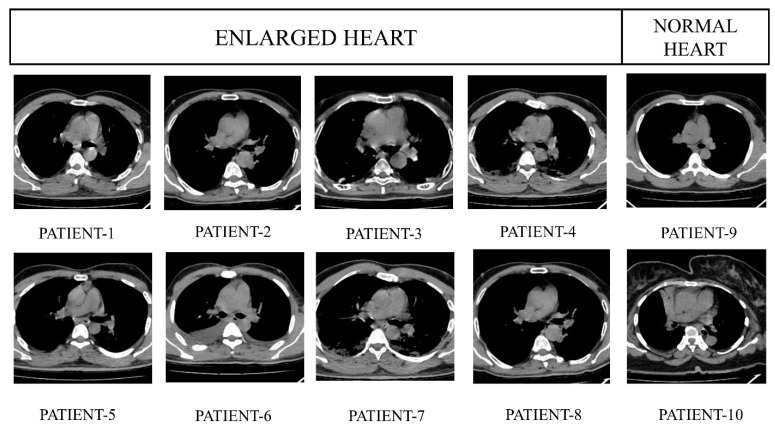
Original images of a healthy heart and enlarged heart.

**Figure 11 jpm-13-01338-f011:**
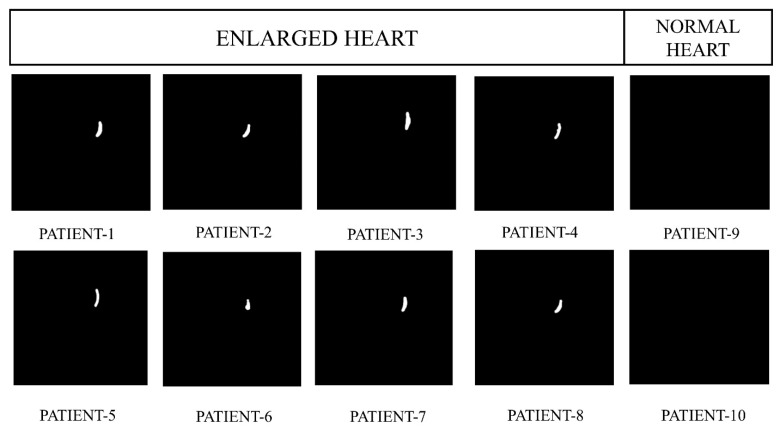
Labeled images of a healthy heart and enlarged heart.

**Figure 12 jpm-13-01338-f012:**
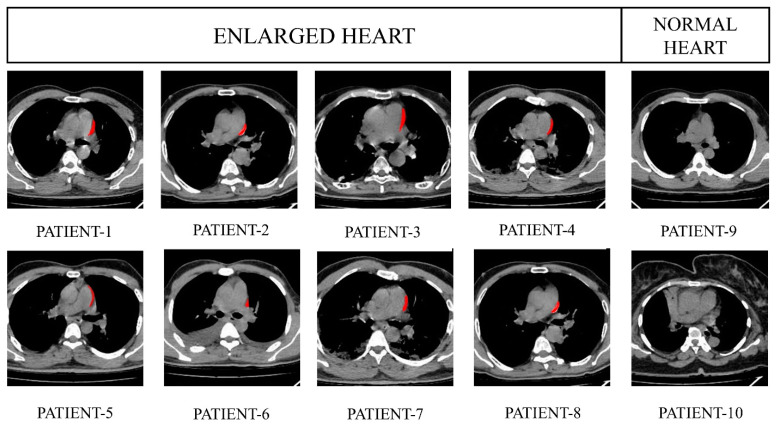
Output images of a healthy heart and enlarged heart.

**Figure 13 jpm-13-01338-f013:**
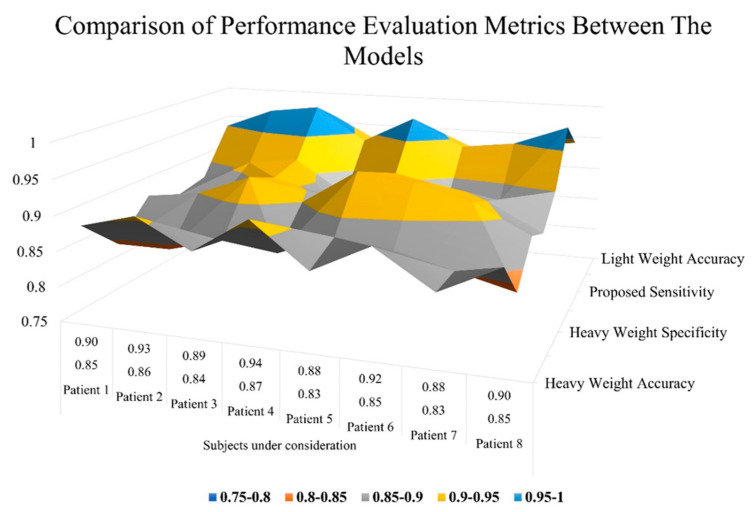
Graphical representation for performance evaluation metrics of models.

**Figure 14 jpm-13-01338-f014:**
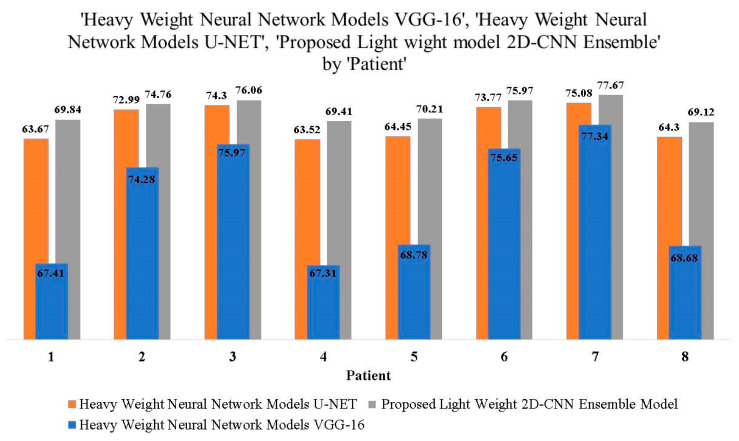
Dice value outcomes of neural network models.

**Figure 15 jpm-13-01338-f015:**
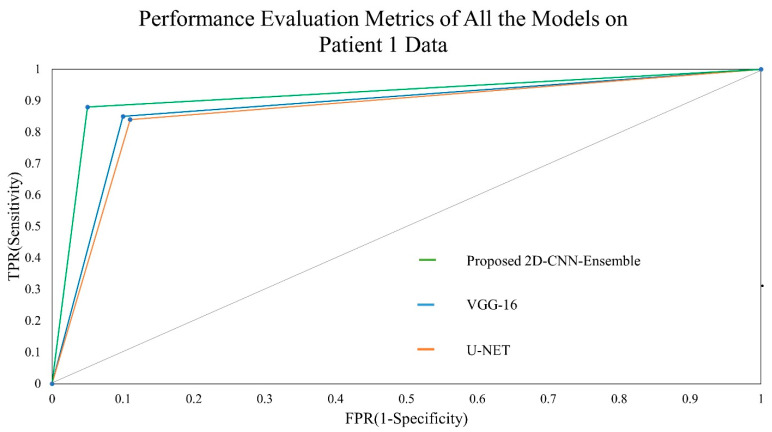
ROC curve between heavyweight and lightweight convolutions of one sample subject.

**Table 1 jpm-13-01338-t001:** Comparison of training time between different models.

Evaluation Parameter	VGG-16	U-Net	Proposed Tiny-2D-CNN-Ensemble
Training Time (in Hours)	13	24	5

**Table 2 jpm-13-01338-t002:** Comparison of Performance Evaluation Metrics between the models.

Model	Performance Evaluation Metrics	Subjects under Consideration							
		Patient 1	Patient 2	Patient 3	Patient 4	Patient 5	Patient 6	Patient 7	Patient 8
U-NET Heavyweight	Sensitivity	0.85	0.86	0.84	0.87	0.83	0.85	0.83	0.85
	Specificity	0.90	0.93	0.89	0.94	0.88	0.92	0.88	0.90
	Accuracy	0.89	0.91	0.87	0.92	0.86	0.90	0.85	0.89
VGG-16 Heavyweight	Sensitivity	0.84	0.84	0.86	0.85	0.88	0.87	0.85	0.83
	Specificity	0.89	0.90	0.93	0.90	0.95	0.94	0.92	0.88
	Accuracy	0.87	0.88	0.91	0.89	0.93	0.92	0.90	0.86
Proposed 2D-CNN-Ensemble Lightweight	Sensitivity	0.88	0.91	0.92	0.87	0.91	0.87	0.88	0.90
	Specificity	0.95	0.98	0.99	0.94	0.98	0.94	0.95	0.98
	Accuracy	0.93	0.95	0.96	0.92	0.95	0.92	0.93	0.94

**Table 3 jpm-13-01338-t003:** Comparison of Dice Coefficient Values in Percentage.

Patient	Heavyweight Neural Network Models		Proposed Lightweight Model
	VGG-16	U-NET	Tiny-2D-CNN-Ensemble
1	67.41	63.67	69.84
2	74.28	72.99	74.76
3	75.97	74.30	76.06
4	67.31	63.52	69.41
5	68.78	64.45	70.21
6	75.65	73.77	75.97
7	77.34	75.08	77.67
8	68.68	64.30	69.12

## Data Availability

The dataset supporting the conclusions of this article is available in the Mendeley Data repository, Chest-CT Images Dataset, doi: 10.17632/w4mv8ypsr3, accessed on 10 February 2023.

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
