# Peer review of "An Efficient, Lightweight, Tiny 2D-CNN Ensemble Model to Detect Cardiomegaly in Heart CT Images"

_jpm, 2023, doi:10.3390/jpm13091338_

Round 1

Reviewer 1 Report

In the study by Ali Al Bataineh, they investigated “An Efficient Tiny Light Weight 2D-CNN-Ensemble Model to Detect Cardiomegaly in Heart CT Images.”.

Image diagnosis have very high affinity for deep learning, and the development of this field has been remarkable. However, many physicians do not understand deep learning, thus your article with review elements has important significance. However, there seem to be many defects in your article.

The purpose of this study was unclear. In this paper, I did not understand what the hypothesis was and what you want to clarify finally.

You are concerned about the overlook of LVH in low-middle income countries in the abstract and introduction. Is the purpose to reduce the cost of finding LVH? If so, this method that requires CT is not a very effective means. This is because screening for LVH by electrocardiogram already achieves high specificity.

The results show that the Tiny-2D-CNN-Ensemble model has a shorter training time than the hjeavy weight U-Net or VGG16.

was the purpose to compare the training time?

In addition, the meaning of finding LVH from CT is not clear. LVH and sudden death are related, however can LVH detection predict sudden death? For example, epidemiologically, the prognosis of apical hypertrophy is relatively good. There seems to be some leap in your logic. Please describe the relationship between LVH detection and sudden death.

It was also unclear whether this paper is for AI researchers or physicians. If this paper has a meaning of review for image diagnosis by AI for physicians, it was not successful in because there are many complicated explanations.

Many physicians do not understand AI such as deep learning, perceptron, or CNN. They probably can’t understand the difference between U-Net and VGG16. You have spent a lot of text on U-net, VGG16, and your ensemble methods, it will limit the readership.

Thus, I suggest you rewrite them more simply.

Author Response

Thank you for your valuable feedback and comments.

Reviewer 2 Report

This interesting study may lead to an ulterior diagnostic tool for cardiopathy. Computer-aided diagnostic tools are more and more developed and will have a role in clinical practice, particularly in resource-limited contexts where they may facilitate and speed the diagnosis process and treatment. Systems currently available are not always compatible with the diagnosis purpose, they are built on complex technologies, and they need good quality basic datasets which may influence training and processing time and then their efficacy. Probably further studies are needed to 

Need some minor editing

Author Response

Thank you for your valuable feedback.

Reviewer 3 Report

The authors throught this study aim to address the challenge of late diagnosis of Cardiomegaly, especially in resource-limited regions where there is a shortage of radiography equipment and specialists. Their primary objective is to develop a Lightweight, Tiny 2D-CNN-Ensemble model for automated screening. By leveraging deep convolutional neural networks (CNN), they hope to provide a solution that can aid in earlier detection of the condition. 

It has been well studied however the presentation currently is lacking and needs major revamp

For example, 

Introduction – is too long and contains parts of discussion. The general statement of CVD is not needed and also “blood vessel damage” is layman term not suited to scienetic literature. 

Would focus on the problem that is cardiomegaly, usual ways to detect and then introduce deep learning. What is currently known in the literature and what are the gaps and how the study aims to addresss these. Finally, explicitly state the aims of the study.

Subheadings such as related works is not needed in introduction which should not be longer than 1-1.5 page max. Much of it needs to be discussed in the discussion

It appears as if Methods and results have been combined and there is no distinction. 

Further, after it there needs to be section on discussion which will combine everything and discuss the results – new findings/strength and limitations of study/comparison with other similar studies. 

Needs further review and less use of lay terminology like blood vessel damage 

Author Response

(The authors gave the same response as above.)

Round 2

Reviewer 1 Report

The authors have addressed essentially all my previous comments, and their revisions have substantially improved the manuscript. I have no further comments.

Author Response

Thank you so much!!